# Ras/ERK and PI3K/AKT signaling differentially regulate oncogenic ERG mediated transcription in prostate cells

**Brady G. Strittmatter** [1], **Travis J. Jerde** [2], **Peter C. Hollenhorst** [3] *

**1** Department of Molecular and Cellular Biochemistry, Indiana University, Bloomington, Indiana, United States of America, **2** Department of Pharmacology and Toxicology, Indiana University School of Medicine, Indianapolis, Indiana, United States of America, **3** Medical Sciences, Indiana University School of Medicine, Bloomington, Indiana, United States of America

* pchollen@iu.edu

**Data Availability Statement:** All RNA-seq and ChIP-seq files are available in the GEO database under accession number GSE164859.

## Abstract

The *TMPRSS2/ERG* gene rearrangement occurs in 50% of prostate tumors and results in expression of the transcription factor ERG, which is normally silent in prostate cells. ERG expression promotes prostate tumor formation and luminal epithelial cell fates when combined with PI3K/AKT pathway activation, however the mechanism of synergy is not known. In contrast to luminal fates, expression of ERG alone in immortalized normal prostate epithelial cells promotes cell migration and epithelial to mesenchymal transition (EMT). Migration requires ERG serine 96 phosphorylation via endogenous Ras/ERK signaling. We found that a phosphomimetic mutant, S96E ERG, drove tumor formation and clonogenic survival without activated AKT. S96 was only phosphorylated on nuclear ERG, and differential recruitment of ERK to a subset of ERG-bound chromatin associated with ERG-activated, but not ERG-repressed genes. S96E did not alter ERG genomic binding, but caused a loss of ERG-mediated repression, EZH2 binding and H3K27 methylation. In contrast, AKT activation altered the ERG cistrome and promoted expression of luminal cell fate genes. These data suggest that, depending on AKT status, ERG can promote either luminal or EMT transcription programs, but ERG can promote tumorigenesis independent of these cell fates and tumorigenesis requires only the transcriptional activation function.

## Author summary

*ERG* is the most common oncogene in prostate cancer. The ERG protein can bind DNA and can activate some genes and repress others. Previous studies indicated that ERG cannot promote cancer by itself, but that ERG works together with mutations that activate the protein AKT. In this study we found that activation of AKT changes the genes that ERG regulates, leading to luminal epithelial differentiation, which is a hallmark of most prostate tumors. However, we also found that a mutant version of ERG that can activate, but cannot repress genes, can drive prostate tumorigenesis without activation of AKT, but this mutant ERG cannot promote luminal differentiation. Our findings suggest that ERG

**Funding:** This work was supported by the National Cancer Institute (www.cancer.gov) of the National Institutes of Health under Award Number R01CA204121 (P.C.H.), and with support from the National Institute of Health (T32 GM131994) the College of Arts and Sciences of Indiana University – Bloomington and the University Graduate School (B.G.S.). The funders had no role in study design, data collection and analysis, decision to publish, or preparation of the manuscript.

**Competing interests:** The authors have declared that no competing interests exist.

mediated tumorigenesis only requires ERG's activation function and can occur independent of luminal cell differentiation.

## Introduction

Prostate cancer is the most common, and second deadliest, malignancy amongst American men [1]. In ~50% of prostate cancers a chromosomal rearrangement results in the fusion of the androgen-regulated promoter of *TMPRSS2* to the open reading frame of *ERG* and results in aberrant expression of either full-length, or N-terminally truncated ERG protein in prostate epithelium [2]. ERG is an ETS family transcription factor that is not expressed in normal prostate epithelial cells [3]. In mouse models, ERG expression cooperates with mutations that activate the PI3K/AKT pathway, such as PTEN deletion, to drive prostate adenocarcinoma [4,5,6]. Consistent with this two-hit model, PTEN deletion positively correlates with *TMRPSS2/ERG* rearrangement in patient tumors [7]. However, the mechanism of cooperation between ERG and the PI3K/AKT pathway is not known. It is also unclear which transcriptional targets of ERG are necessary to promote tumor formation. ERG can both activate and repress the expression of direct target genes, but the relative role of these activities in tumor formation is also in question.

ERG expression can influence prostate cell fate decisions. The epithelium of the normal prostate gland is composed primarily of luminal secretory cells separated from the stroma by a layer of supportive basal epithelial cells. Most prostate tumors are characterized by an expansion of luminal epithelial cells and a relative absence of basal epithelial cells [8]. ERG expression in prostate tumor models can promote luminal epithelial cell fates, indicating that this could be part of the oncogenic function of ERG [8,9,10]. The androgen receptor (AR) is a key transcription factor that promotes luminal cell fates and AR activity is necessary for the growth of all early stage prostate tumors. Several studies indicate that ERG can cooperate with AR in promoting luminal cell fate [11,12,13]. In contrast to these findings, we and others have shown that expression of ERG in the immortalized-normal prostate cell line RWPE1 does not promote luminal epithelial differentiation, but rather promotes the seemingly opposite phenotypes of migration, invasion, and epithelial to mesenchymal transition (EMT) [14,15,16,17,18]. Despite this difference regarding cell fate, we have found that ERG expression and PI3K/AKT activation cooperate to promote RWPE1 xenograft tumor growth [19]. Therefore, with regard to tumor formation, ERG expression has a similar role in RWPE1 xenografts and transgenic mouse models.

ERG can either activate or repress target genes. Studies that have measured expression changes of direct ERG target genes upon addition or depletion of ERG in prostate cells have identified approximately equal numbers of activated and repressed genes [10,20,21]. A recent study [8] indicates that direct ERG repression of TP63, a master regulator of basal cell fate, is important for ERG to promote luminal fates. ERG has been demonstrated to downregulate *VCL*, an epithelial related cytoskeleton protein [22] as well as regulators of the PI3K/AKT pathway *PTEN* and *IRS2* [23,24]. ERG interacts with various co-repressors including EZH2, HDAC1, HDAC2, and KDM4A [22,25]. Transcriptional activation by ERG is thought to be important for oncogenic function. ERG can activate *CTGF*, *ENC1*, *ETS1*, *PLAU*, and *VIM* which are all thought to be involved promoting oncogenic phenotypes [15,26,27] and has been shown to interact with co-activators P300, BRD4, and the BAF Complex [9,28]. We have also identified an interaction with the co-activator EWS that is necessary for ERG to promote tumorigenesis in the RWPE1 xenograft system [19].

Both the Ras/ERK and PI3K/AKT signaling pathways are critical regulators of ERG function in prostate [29]. ERK2 can phosphorylate ERG at S96 and S215 in prostate and endothelial cells as well as at S276 in leukemic cells [21,30,31]. ERG gene rearrangements can result in expression of either full-length ERG, or N-terminal deletions of 32 or 92 amino acids (numbering is Uniprot isoform 1, isoform 2 adds seven amino acids) and S96 is phosphorylated in all three of these proteins [21]. Phosphorylation of ERG S215 by ERK results in a conformational change in the ERG protein that allows subsequent ERK phosphorylation at ERG S96. S96 phosphorylation allows for transcriptional activation by disrupting the interaction between ERG and the Polycomb Repressive Complex 2 (PRC2) [21]. The role of the PI3K/AKT pathway in ERG function is less well understood mechanistically. AKT has not been demonstrated to directly phosphorylate ERG and the canonical kinase complexes downstream of AKT, mTORC1, and mTORC2 are not involved in ERG transcriptional activity [32]. GSK3B, a downstream target of AKT, has been shown to phosphorylate ERG and alter tumorigenic capabilities, but only when DNA damage is present and WEE1 signaling is active [33].

Here, we demonstrate that a phospho-mimetic mutation of ERG at S96 (S96E) can promote clonogenic growth and tumorigenesis in a mouse xenograft model independent of AKT activation. Further, we show that only a portion of cellular ERG is phosphorylated in prostate cells and that this phosphorylation takes places in the nucleus at sites of ERK recruitment. ERK binding to chromatin correlated with ETS/AP-1 motifs and ERG-activated genes. ERG S96E counteracted ERG-mediated repression at sites not bound by ERK by reducing EZH2 binding and depleting H3K27me3. Activation of AKT redistributed ERG to new binding sites and allowed ERG to promote a luminal gene expression program. Consistent with these findings, tumors driven by ERG/AKT expressed AR, while tumors caused by ERG S96E did not. Tumors with both ERG S96E and activated AKT did not express AR, indicating that ERG-mediated repression is required for luminal cell fates, but not for tumorigenesis.

## Results

### The Ras/ERK and PI3K/AKT signaling pathways differentially regulate ERG mediated phenotypes

To determine the roles of the Ras/ERK and PI3K/AKT pathways in ERG-mediated phenotypes, FLAG-tagged WT ERG and ERG S96 phospho-mutants were stably expressed with or without constitutively activate myristoylated-AKT (mAKT) in RWPE1 normal prostate epithelial cells. ERG Isoform 1 was used (Uniprot) in this study. Isoform 1 differs from isoform 2 by the N-terminal amino acids (MAST in isoform 1 and MIQTVPDPAAH in isoform 2). ERG S96 (S103 in isoform 2) is phosphorylated by ERK1/2 and S96E functions as a phosphomimetic, with S96A a phosphonull [21]. Immunoblotting of whole cell lysates confirmed ERG expression and showed an increased level of S473 phosphorylated AKT in the lines expressing mAKT (Fig 1A). Similar to our previous findings [21] ERG and ERG-S96E significantly increased RWPE1 cell migration while ERG-S96A had no effect. AKT activation alone increased RWPE1 migration, but the expression of mAKT in combination with ERG or ERG S96E did not further increase RWPE1 migration (Figs 1B and S1A). The same lines were tested in a clonogenic survival assay (Figs 1C and S1B). ERG promoted a moderate increase in clonogenic survival, and this function was not lost in the ERG S96A mutant. Expression of mAKT alone also moderately increased colony formation, however, the combination of ERG and mAKT significantly and synergistically increased colony formation. Further, expression of ERG-S96E alone promoted colony formation to a similar level as the ERG/mAKT combination, but neither ERG S96E nor ERG S96A synergized with mAKT. Together, these data suggest that S96 phosphorylation, but not AKT activation, is required for ERG to promote cell

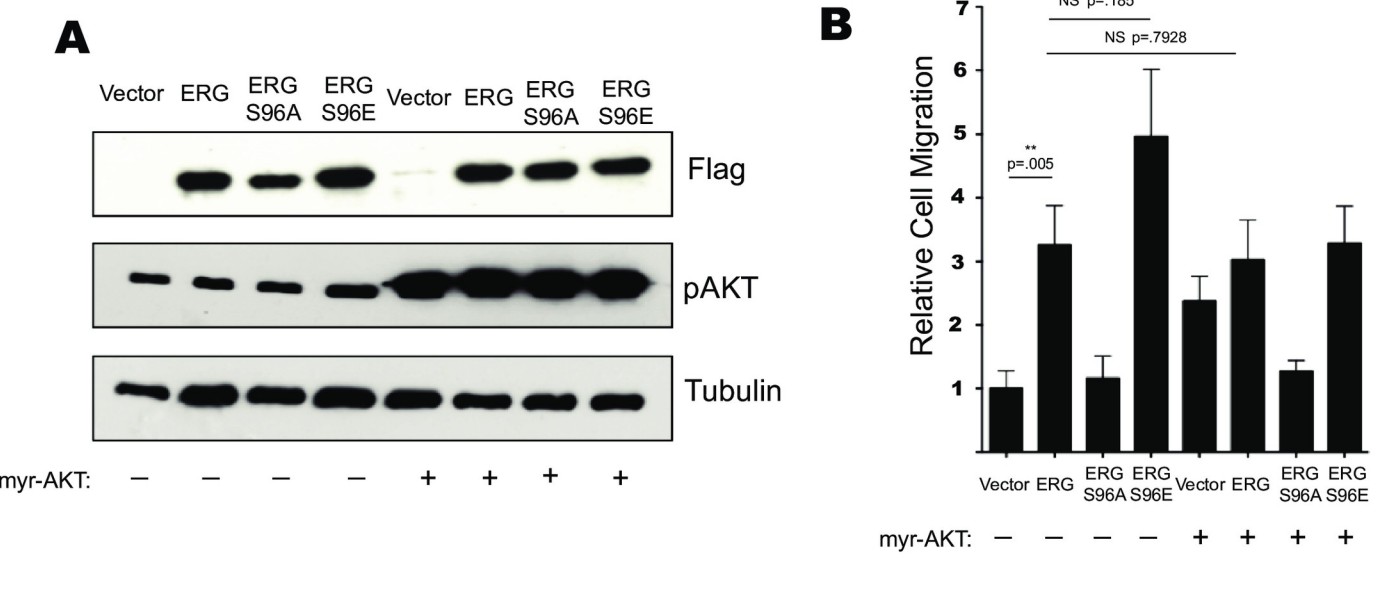

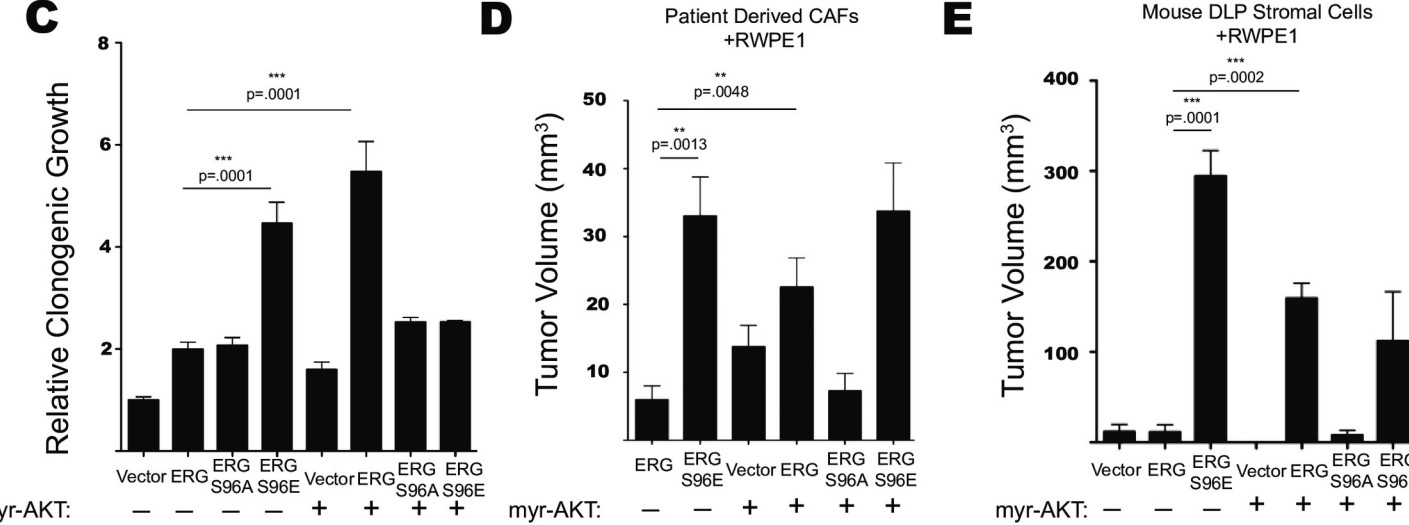

**Fig 1. Ras/ERK and PI3K/AKT signaling differentially regulate ERG mediated phenotypes.** (A) Immunoblot of RWPE1 cells expressing FLAG-tagged ERG or mutant, and myristoylated AKT as indicated. (B) Transwell cell migration of RWPE1 cells expressing indicated constructs. Shown are the mean and SEM normalized to vector of 4 biological replicates, each the mean of 2 technical replicates. (C) Clonogenic growth of RWPE1 cells sparsely plated and allowed to form colonies over 10 days followed by fixing, staining, and quantification via Sygene GeneTools colony counting software. (D) Mouse xenograft experiments measured by calipers of RWPE1 cell lines grown (n = 6) in the flanks of immunocompromised mice for 8 weeks with patient derived CAFs or (E) RWPE1 cells grown (n = 4 or 6) with DLP-derived stromal cells from INK4A null mice.

migration, and either S96 phosphorylation or AKT activation allows ERG to promote clonogenic growth.

The cell lines were then tested in two different mouse xenograft models to assay tumor formation ability. First, RWPE1 cell lines, primary patient derived cancer associated fibroblasts (CAFs), and matrigel were co-injected into the flanks of nude mice and allowed to form tumors for eight weeks (Fig 1D). Consistent with previous findings using this system [19], cells expressing ERG in combination with mAKT formed significantly larger tumors than those expressing ERG or mAKT alone. Similar to the clonogenic growth assay however, the S96E mutation allowed ERG to promote tumor formation in the absence of mAKT. Overall, the

sizes of the tumors grown with CAFs were much smaller than in previous work [19]. We attributed this change to differences in the primary CAF cells. To confirm these results in a more defined system, the same RWPE1 cell lines, and matrigel were co-injected with DLP derived stromal cells from *INK4a* null mice as alternative support cells [34]. This system revealed a similar trend in relative tumor growth, but with larger tumors (Fig 1E). ERG synergized with mAKT to promote tumor formation. Expression of ERG-S96E alone formed the largest tumors and grew statistically significant tumors by week four (S1C Fig). Similar to the clonogenic growth assay, the addition of mAKT actually decreased tumor size when combined with ERG S96E. Overall, our data indicates that the ERG-S96E phosphomimetic mutation abrogates the need for AKT activation in ERG-mediated tumor formation.

## Phosphorylated ERG is in the nucleus

To further investigate how ERK phosphorylation regulates ERG-mediated transcription we conducted an immunoprecipitation (IP) coupled with mass spectrometry of total cellular ERG from VCaP prostate cancer cells which express ERG due to a *TMPRSS2/ERG* gene rearrangement. We identified seven high confidence phosphorylation sites on the ERG protein (Fig 2A) including residues previously identified [21,30] to be phosphorylated by the Ras/ERK pathway including S96, S215, and S276 (S283 in isoform 2). Acetylation of ERG at K89 and K92 was also detected (S2A Fig). These acetylation sites have been demonstrated to play a key role regulating ERG function in Acute Myeloid Leukemia [28]. Eight unique phospho-peptides and twelve unique apo-peptides were identified in the region spanning S96 (Fig 2A) indicating that a portion (~40%) of total cellular ERG is phosphorylated at S96 in VCaP cells.

IP-mass spectrometry of RWPE1 cells expressing ERG (RWPE-ERG) identified three high confidence phosphorylation sites (Figs 2A and S2B). Two of these sites, S96 and S215 were also identified in VCaP. Five apo-peptides and three phospho-peptides (Fig 2A) spanned the S96 region indicating that a similar portion of cellular ERG was phosphorylated at S96 in VCaP and RWPE-ERG cells. In order to identify the cellular location of phosphorylated ERG, nuclear/cytoplasmic fractionation was performed in RWPE1-ERG cells. ERG was found in both the cytoplasmic and nuclear fraction, however only nuclear ERG was phosphorylated at S215 (Fig 2B).

## ERK binds chromatin at sites of ERG activation function

We previously demonstrated [21] that S96E and S96A do not alter nuclear/cytoplasmic ratios of ERG, indicating that phosphorylation does not alter nuclear trafficking. Therefore, to explain the bias of phosphorylated ERG for the nucleus, we postulated that a non-diffusible fraction of ERG might be phosphorylated by ERK in the nucleus. The ETS transcription factor ELK1 is phosphorylated when bound to a subset of target genes via differential recruitment of ERK to chromatin [35]. To determine if ERK similarly associates with a subset of chromatin-bound ERG, we conducted chromatin immunoprecipitation-sequencing (ChIP-Seq) of ERK2 in RWPE-ERG cells. We found ERK2 was present at a portion of ERG binding sites (Fig 2C). To determine if specific sequence motifs were associated with ERK2 occupancy, the top 500 ERG-bound regions enriched by ERK2 ChIP and the bottom 500 ERG-bound regions with no ERK2 ChIP signal were analyzed by RSAT motif using each data set as a control for the other to allow identification of differential motifs. A non-canonical ETS motif and an AP1 motif were significantly enriched at ERG/ERK2 co-bound regions, while ERG-alone regions were associated with CpG and SMAD motifs (Fig 2C). Canonical ETS motifs were enriched in both datasets, and therefore cancelled out. Gene Set Enrichment Analysis (GSEA) was also used to identify enriched motifs by ranking ERK2 ChIP signal across ERG binding sites (Fig 2C). This

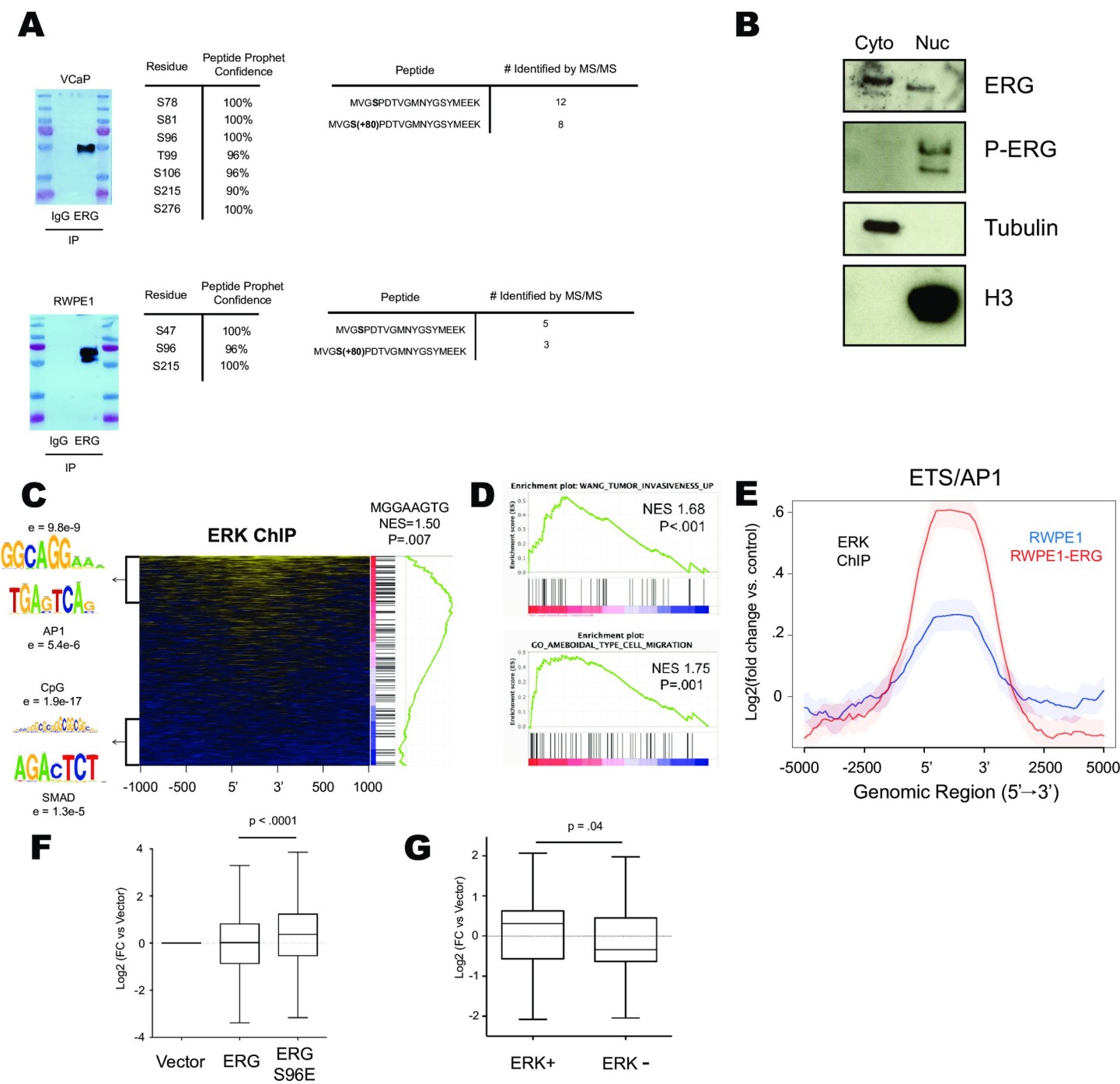

**Fig 2. Ras/ERK signaling occurs at ETS/AP1 genomic locations to regulate cell migration.** (A) MS/MS analysis of ERG immunoprecipitated from VCaP and RWPE-ERG cells. Data tables show phospho-residue number of ERG NCBI Isoform 1 and compares number of apo/phospho peptides with coverage of S96. (B) Cytoplasmic/Nuclear fractionation of RWPE-ERG immunoblotted for indicated proteins. (C) Heatmap of ERK ChIP-Seq data centered on ERG bound regions [15] depicting motifs over-represented in the top 500 ERK enriched and bottom 500 ERK depleted genes and GSEA analysis of ERK enrichment across ERG bound regions run for motif analysis. (D) GSEA analysis of ERK ChIP-Seq data enrichment ranked on ERG bound regions. (E) ERK ChIP-Seq data centered on consensus ETS/AP1 motifs previously found to be ERG bound in RWPE1. (F) Log2(FC) of nearest genes bound by ERG in RWPE1 cells determined by ChIP-Seq in cells expressing ERG or ERG S96E. (G) Log2(FC) of nearest gene to ERG alone or ERG/ERK co-bound regions.

analysis identified a sequence (MGGAAGTG) consistent with the ETS/AP-1 half site spacing (GGAAGTGA) we have previously reported is enriched in the regulatory regions of cell migration genes [36]. Consistent with a cell migration role for ERK/ERG target genes, GSEA of ERK ChIP signal across ERG-bound regions was enriched for the gene ontologies Ameboidal Type Cell Migration and Tumor Invasiveness (Fig 2D).

To further investigate the recruitment of ERK to ETS/AP-1 motifs, we subjected wild-type RWPE1 to ERK2 ChIP-seq (Fig 2E). In the absence of ERG, ERK2 was enriched at the same ETS/AP-1 motifs that ERG binds, indicating that some factor other than ERG does recruit ERK2 to these sites. However, the ERK2 enrichment at ETS/AP-1 motifs was higher in RWPE-ERG cells (Fig 2E) and this enrichment was not observed at ERK binding sites that lack ERG (S2D Fig), indicating that ERG increases ERK2 recruitment. This is consistent with our previous finding that ERG directly binds ERK with relatively high affinity compared to other ETS factors [32].

To test how ERK binding and phosphorylation of ERG alters ERG mediated transcription, we analyzed previously published RNA-Seq data from RWPE-vector, RWPE-ERG and RWPE-ERG S96E cell lines (GSE86232) [21]. Previous studies indicate that ERG can activate some genes and repress others [10,20,22]. Consistent with this, we found that genes near ERG binding sites were both activated and repressed resulting in a mean expression change of zero when ERG is expressed in RWPE1 cells (Fig 2F). In contrast, expression of these genes in cells expressing ERG-S96E was significantly higher than in RWPE-vector cells (Fig 2F), suggesting S96 phosphorylation favors gene activation. We then tested the correlation with ERK1 binding. Genes near ERK-ERG co-bound sites were activated, while genes near ERG bound sites lacking ERK were repressed (Fig 2G), suggesting that ERK binding allows ERG to switch from a repressor to an activator.

## Ras/ERK signaling opposes PRC2 repression at ERG binding sites

Phosphorylation of ERG at S96 by ERK disrupts ERG's interaction with polycomb repressive complex 2 (PRC2) including the enzymatic subunit EZH2 [21]. To further investigate the interplay between ERG, ERK, and EZH2 on chromatin we analyzed previously published ChIP-Seq data of ERG and EZH2 in RWPE1, RWPE1-ERG, and RWPE1-ERG S96E cells (GSE86232) and conducted ChIP-Seq of H3K27me3, the repressive mark that EZH2 deposits on chromatin. We examined occupancy at ERG-bound regions in promoters or enhancers (Fig 3A). Expression of ERG increased ERK binding, decreased EZH2 binding, and decreased H3K27me3. Loss of EZH2 binding occurred mostly at promoters, but loss of H3K27me3 was observed at both promoters and enhancers. Expression of ERG-S96E resulted in an even further depletion of EZH2 and H3K27me3 at promoters and enhancers compared to ERG alone (Fig 3A). Further, GSEA of ranked ERK binding across ERG-bound regions showed negative correlation with high density CpG promoters marked with H3K27me3 and positive correlation with genes that are upregulated upon knockdown of EZH2 (Fig 3B). These data suggest opposing functions of ERK and EZH2 at ERG binding sites.

ERG S96E can promote tumor formation in the absence of activated AKT signaling, whereas ERG cannot (Fig 1E). To determine how the S96E point mutation alters gene expression, we generated a volcano plot comparing differentially regulated genes between ERG and ERG S96E and found that, consistent with Fig 2F, the majority (77.3%) of the differentially regulated genes were activated in ERG S96E compared to ERG alone (Figs 3C and S3A). This ability of ERG S96E could be due to increased activation of activated targets, loss of repression of repressed targets, or both. Therefore, we compared differences in the ERG-activated targets co-bound by ERK2 and the ERG-repressed targets bound by ERG alone (Fig 3D). ERG S96E

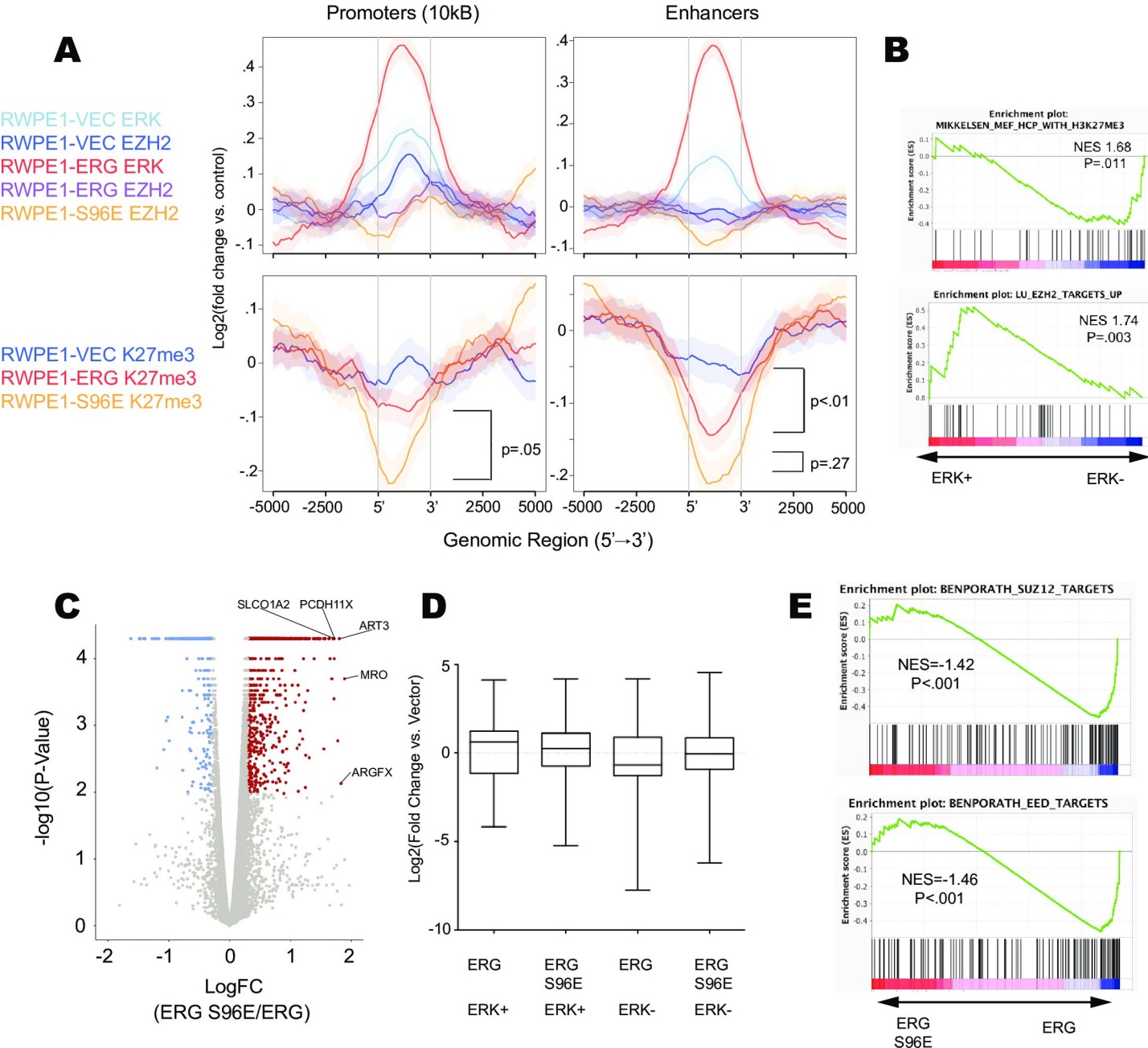

**Fig 3. Ras/ERK signaling opposes PRC2 repression at ERG binding sites.** (A) Average plots of ERK, EZH2, and H3K27me3 ChIP-Seq conducted in RWPE1, RWPE1-ERG, or RWPE1-ERG S96E cell lines centered on ERG bound regions in RWPE1 cells [15] (B) GSEA analysis of ERK ChIP signal ranked across ERG bound regions in RWPE1 cells. (C) Volcano plot showing differentially regulated genes determined by RNA-Seq between RWPE1 cell lines expressing ERG S96E and ERG (red— significantly activated / blue significantly repressed). (D) Log2(Fold change vs. Vector) of genes nearest ERG-alone or ERG/ERK co-bound regions in RWPE1 cells expressing ERG or ERG S96E. (E) GSEA analysis of differentially regulated genes called by RNA-Seq comparing RWPE1 ERG S96E to RWPE1 ERG.

did not further activate ERG/ERK co-bound target genes, consistent with these genes already being associated with phosphorylated ERG. Instead the major effect of ERG S96E was loss of repression of genes bound by ERG alone. Further, GSEA analysis of the differentially regulated genes between ERG S96E and ERG found that ERG S96E was negatively correlated with targets of EED and SUZ12, two essential components of the PRC2 (Fig 3E). Together these data

suggest that the major role of the S96E mutation is the loss of ERG-mediated repression via PRC2 recruitment.

## Constitutive PI3K/AKT signaling allows new ERG binding across the genome

Wild-type ERG cannot promote tumor formation alone, but ERG can promote prostate tumor formation in cooperation with mutations that activate the PI3K/AKT pathway (Fig 1E) [4,10,37]. The molecular mechanism by which ERG and AKT synergize to promote tumor formation is not known. To investigate this, ChIP-seq of ERG in the presence of mAKT was compared to ChIP-seq of ERG alone and ERG S96E alone. As we have previously reported [21], S96E did not significantly alter ERG binding (Fig 4A, bottom). In the presence of mAKT, ERG maintained these same binding sites (Fig 4A, bottom), but gained new binding sites (Fig 4A, top). Further comparison with an additional ERG ChIP-seq in RWPE-ERG cells [38] again showed a common set of binding sites for ERG and ERG S96E, but new ERG binding sites in the presence of mAKT (S4A Fig). To ensure that this difference was not due to experimental variation, a second ERG/mAKT ChIP-seq was performed. By principal component analysis, both replicates of ERG ChIP-seq in cells with mAKT clustered closely together and were distinct from ERG-alone and ERG S96E-alone ChIP-seq (Fig 4B).

Motif analysis of ERG-bound regions in RWPE-ERG/mAKT cells revealed a canonical ETS motif as well as motifs not identified in RWPE1 ERG ChIP-Seq including GGAA repeats that are targets of EWS/FLI1, and FOX/ETS motifs (Fig 4C). Gene ontology analysis conducted on the nearest genes to each ERG peak in RWPE1 ERG/mAKT cells found the highest association with blood vessel development (Fig 4D). This is consistent with the normal function of ERG in endothelial cells, where it is a key regulator of vascular stability and angiogenesis [39].

Differential ERG binding in the presence of mAKT could be due to an alteration of chromatin structure that allows access to novel ETS binding motifs. To test this, ATAC-Sequencing was conducted in RWPE1 cells expressing ERG, mAKT, or both, and revealed that open chromatin regions were strikingly similar in all of the cell lines (Fig 4E, top). In addition, the regions uniquely bound by ERG in the presence of mAKT exhibited similar open chromatin profiles (Fig 4E, bottom). Further, individual sites of emergent ERG binding showed no difference in open chromatin in the presence or absence of mAKT (Fig 4F). These data suggest that the presence of active AKT changes the ERG cistrome without altering chromatin accessibility.

To investigate if AKT inhibition altered ERG binding in a cell line with an endogenous *TMPRSS2-ERG* re-arrangement, ChIP-Seq of ERG was conducted in VCaP cells that were treated for one week with a PI3K inhibitor (1 μM LY294002). PI3K inhibition resulted in a change in the ERG cistrome (S4B Fig). The ERG bound regions in untreated VCaP were near genes enriched for epithelial ontologies, but epithelial ontologies were not enriched among genes near ERG binding sites in cells with PI3K inhibition (S4C Fig). Genes with nearby ERG binding sites in untreated, but not PI3K inhibited, VCaP cells included several markers of epithelial differentiation such as *FEM1B*, *LGR6*, and *FN1* (S4D Fig).

## Constitutive PI3K/AKT signaling co-operates with ERG to promote luminal fate associated gene programs

To determine how ERG, mAKT, and the combination effect gene expression we conducted RNA-Sequencing of these cell lines in triplicate. Expression of mAKT alone resulted in a relatively small number (1,245) of significantly differentially regulated genes (S5 Fig). The addition of ERG resulted in a more robust change in differentially regulated genes (3,367) compared to RWPE1-vector. However, the largest number of differentially regulated genes (3684) was

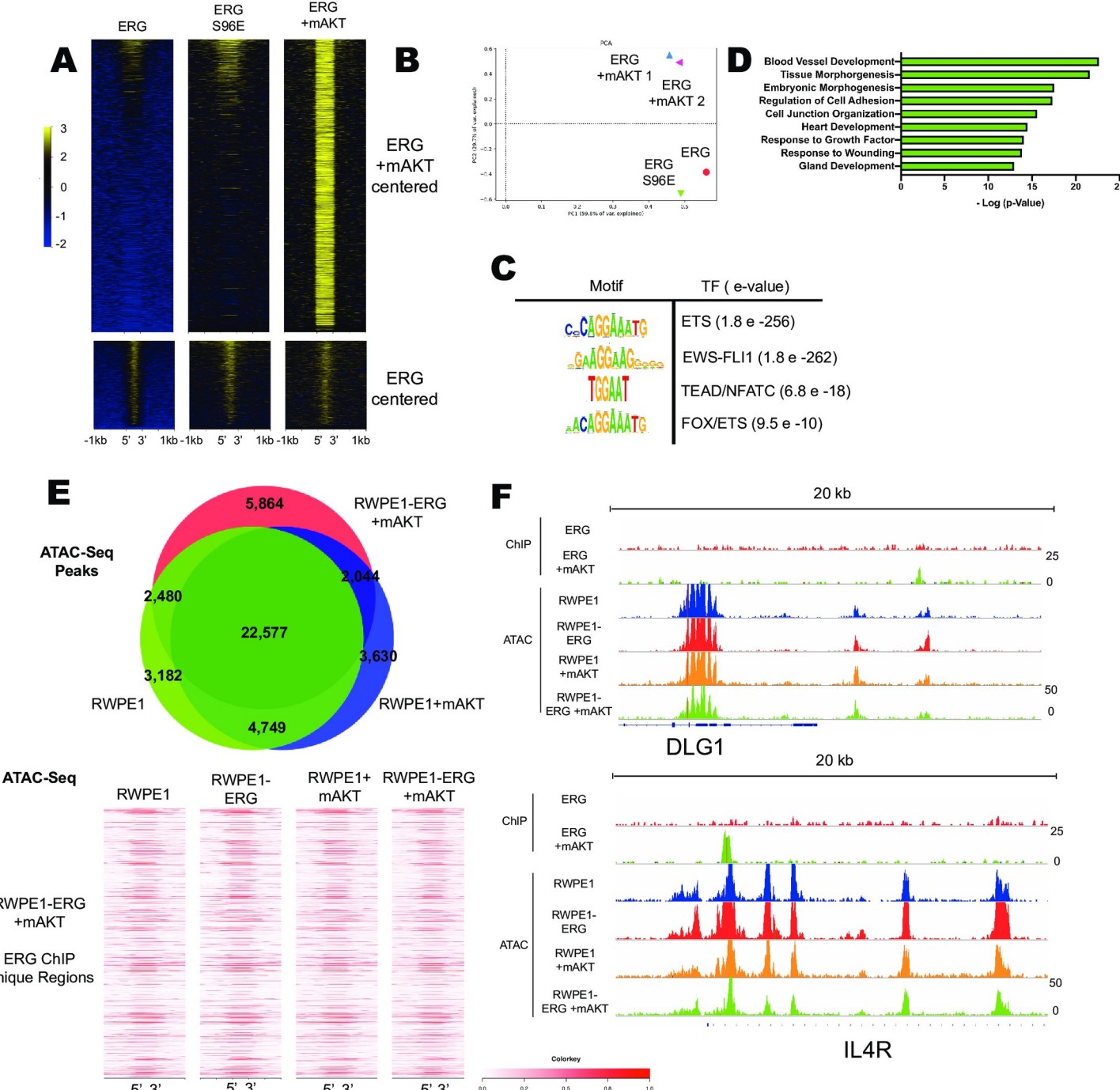

**Fig 4. Constitutive PI3K/AKT signaling allows new ERG binding across the genome.** (A) ChIP-Seq of ERG in RWPE1 cell lines centered on called peaks in RWPE1-ERG/mAKT cell line (top) and centered on peaks called in RWPE1-ERG cells (bottom) (B) PCA analysis of ERG ChIP-Seq with two biological replicates from RWPE-ERG/mAKT. (C) Motif and (D) Gene Ontology analysis of ERG called peaks in RWPE1-ERG/mAKT cells. (E) Overlap of the top 33,000 ATAC-Seq peaks from RWPE1, RWPE1+mAKT, and RWPE1-ERG/mAKT cells and heatmap of ATAC-Seq reads centered on unique peaks in ERG ChIP-Seq from RWPE1-ERG/mAKT cells. (F) Gene tracks of ERG ChIP-Seq and ATAC-Seq from RWPE1 cells depicting two representative regions bound by ERG only in RWPE-ERG/mAKT.

observed when RWPE-ERG/mAKT was compared to RWPE-ERG. This larger role for mAKT in the presence of ERG suggests synergy. Unsupervised clustering of RNA-Sequencing data resulted in two distinct clusters one containing RWPE and RWPE-ERG and the other RWPE-

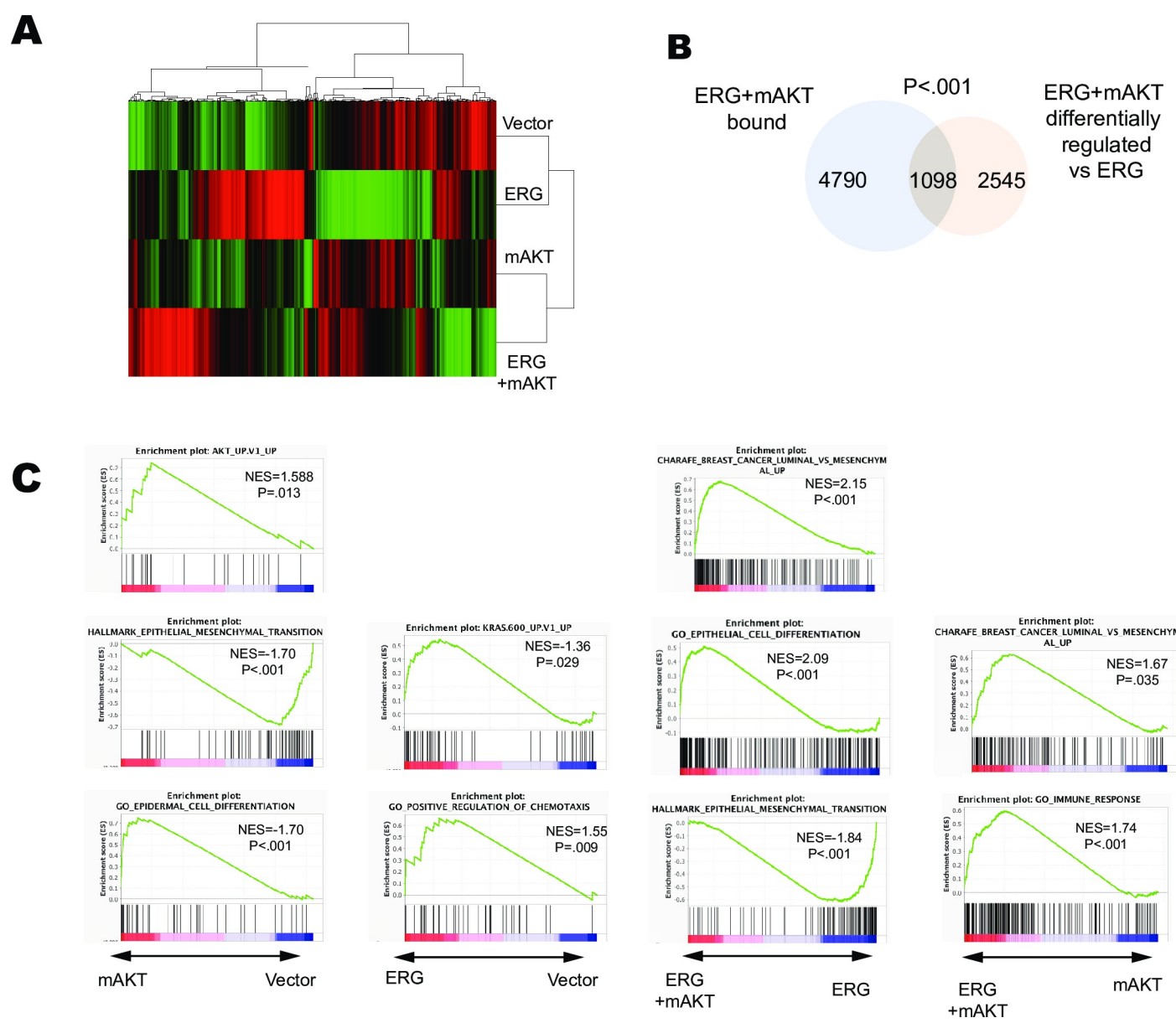

**Fig 5. Constitutive PI3K/AKT signaling co-operates with ERG to promote luminal fate associated gene programs.** (A) Unsupervised clustering of RNA-Sequencing data from RWPE1 cells expressing Vector, ERG, myr-AKT, or ERG+mAKT (green-activated, red-repressed) (B) Overlap of genes nearest ERG bound regions in RWPE-ERG/mAKT and significantly differentially regulated genes in RWPE-ERG/mAKT compared to RWPE/ERG (C) Top GSEA enrichments when comparing gene expression in the RWPE1 cells expressing the indicated constructs (below).

mAKT and RWPE-ERG/mAKT (Fig 5A). Of the genes which were differentially expressed upon addition of mAKT to RWPE-ERG cells, 1098 (30%) had neighboring ERG bound regions in the RWPE-ERG/mAKT ChIP-seq indicating that they are potential direct targets (Fig 5B). This was a statistically significant enrichment (p < 0.001).

GSEA indicated that the addition of mAKT alone to RWPE1 cells resulted in an active AKT signature, inhibition of EMT, and epidermal differentiation (Fig 5C). In contrast, ERG expression alone in RWPE1 cells resulted in an active Ras/ERK signature and was correlated with positive regulation of chemotaxis (Fig 5C), consistent with the ability of ERG to promote cell migration (Fig 1B) and previous findings that ERG expression in RWPE1 can mimic Ras/ERK

signaling [15]. Strikingly, the combination of ERG and mAKT expression in RWPE1 cells promoted a luminal epithelial differentiation program compared to expression of either ERG or mAKT alone. Comparing gene expression from RWPE-ERG/mAKT to RWPE-ERG revealed that the addition of mAKT promoted epithelial and luminal differentiation and inhibited EMT (Fig 5C). Similarly, when comparing RWPE1-mAKT cells to RWPE-ERG/mAKT, the addition of ERG activated programs associated with luminal differentiation and immune response (Fig 5C). Taken together, these data indicate that expression of ERG alone promotes an active Ras/ERK signature responsible for cell migration and EMT, but constitutive activation of the PI3K/AKT pathway reprograms the ERG cistrome and transcriptome to promote luminal epithelial differentiation.

### Ras/ERK and PI3K/AKT pathway regulation of ERG result in distinct gene expression programs in prostate cells

In contrast to ERG alone, both ERG S96E and ERG/mAKT can promote tumor formation (Fig 1E). To determine if ERG S96E and ERG/mAKT commonly regulate an oncogenic gene expression program, RNA-Seq from RWPE-ERG S96E was compared to RWPE-ERG/mAKT and RWPE-ERG. In contrast to our hypothesis that ERG S96E and ERG/mAKT would have a similar role, we found that the majority (62.2%) of the significantly regulated genes in the tumorigenic lines were differentially regulated whereas only 300 genes were regulated in the same direction compared to the non-tumorigenic RWPE-ERG cells. Commonly regulated genes were found to be enriched for Regulation of the MAPK cascade, Cell adhesion, Blood vessel development, as well as KEGG Pathways in Cancer by gene ontology analysis (Fig 6A). In contrast, enriched gene ontologies of differentially regulated genes included B-cell differentiation, neuron differentiation and KEGG FoxO signaling pathway. Interestingly, ERG is implicated several of these processes as part of its normal function and thus differences in Ras/ERK or PI3K/AKT signaling may dictate ERG function both when aberrantly expressed in prostate cancer and in its normal physiological role.

The RWPE1 cell line expresses both luminal and basal cytokeratins and does not express androgen receptor (AR) [40]. Our data suggests that activation of the PI3K/AKT pathway alters the ERG cistrome (Fig 4A and 4B), allowing ERG to promote luminal epithelial cell fates (Fig 5). Immunoblots of the xenograft tumors formed by RWPE-ERG/mAKT cells showed expression of the luminal cell markers AR and FOXA1 and loss of basal marker TP63 (Fig 6B and 6C). In contrast, ERG S96E expression did not alter the ERG cistrome (Fig 4A and 4B) and the tumors formed by RWPE-ERG S96E do not express AR or FOXA1 and maintained TP63 expression (Fig 6B and 6C). Further, the small tumors expressing ERG S96A also did not express AR or FOXA1. This indicates that ERG promotion of androgen receptor expression requires both transcriptional activation and repression functions. Interestingly, ERG S96E tumors had high TP63 expression and ERG S96A tumors had no TP63 expression, consistent with a direct role for ERG in the repression of the *TP63* gene [28]. Immunohistochemistry of the tumor samples from RWPE1-ERG S96E and RWPE1-ERG/mAKT was conducted and the tumors showed strikingly different morphologies (Fig 6D). Both tumors exhibited epithelial characteristics as indicated by Pan-CK staining, however, only RWPE1-ERG S96E maintained strong expression of basal marker KRT5. These findings indicate that ERG/mAKT mediates a loss of basal cell characteristics and a gain of androgen receptor expression in prostate tumors.

### Discussion

This study sheds light on the molecular mechanisms of PI3K/AKT and RAS/ERK regulation of ERG function in prostate cancer. Our findings indicate that ERK is recruited to a subset of

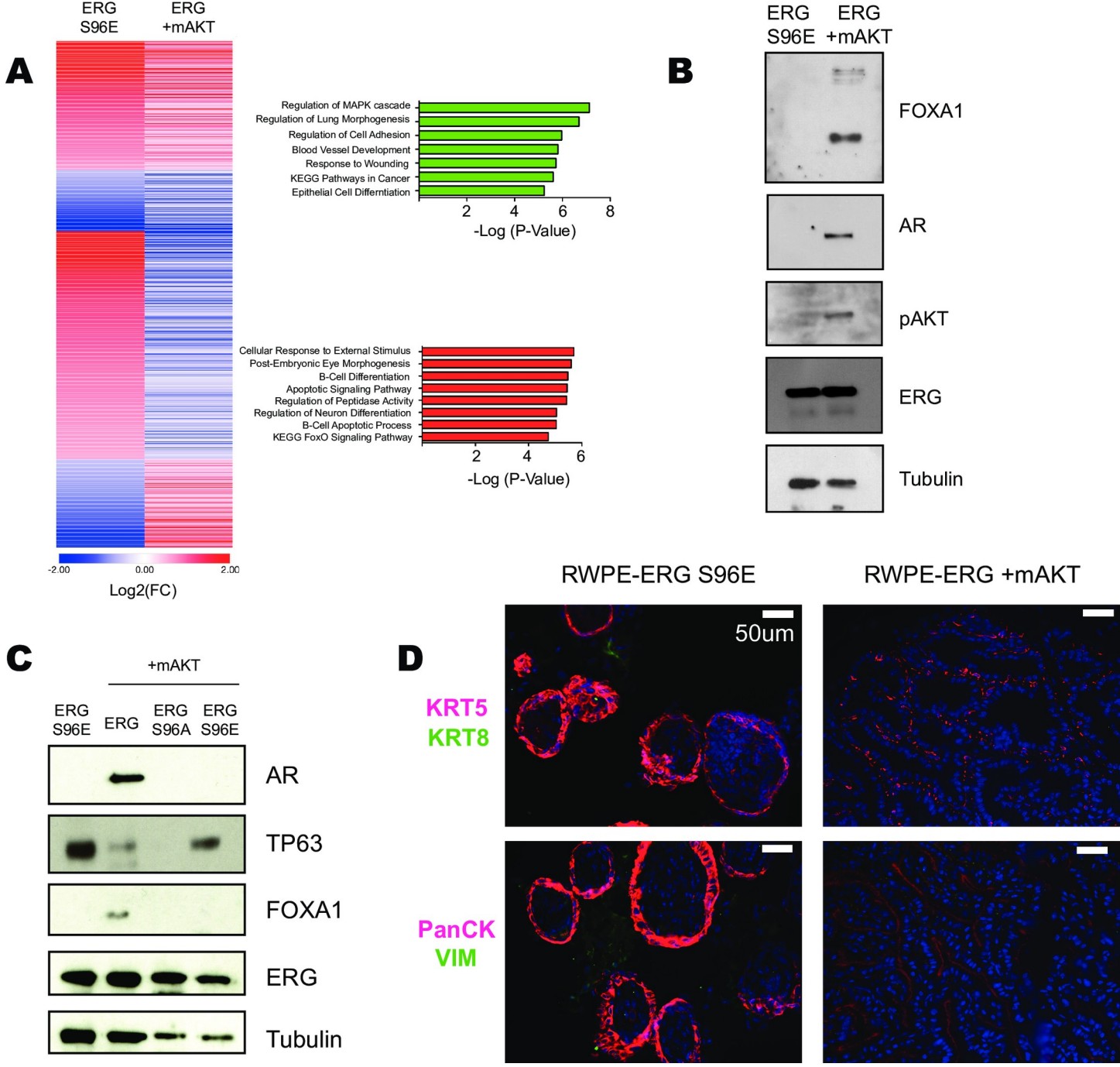

**Fig 6. Ras/ERK and PI3K/AKT pathway regulation of ERG result in distinct gene expression programs in prostate cells.** (A) Heat map comparison of genes in RWPE1-ERG S96E or RWPE1-ERG/mAKT significantly differentially regulated compared to RWPE1-ERG alone. Gene ontology analysis of co-regulated genes (green) and differentially regulated genes (red). (B) Immunoblot of RWPE1 xenograft tumor lysates comparing those expressing only ERG-S96E and those expressing ERG and mAKT. (C) Immunoblot of RWPE1 xenograft tumor lysates expressing phospho-mutants of ERG. (D) Histology of RWPE1-ERG S96E or RWPE1-ERG/mAKT xenograft tumors. Top images stained for KRT5 (basal) and KRT8 (luminal) markers and bottom images stained for Pan-CK and VIM.

genomic ERG targets where it phosphorylates ERG, allowing for transcriptional activation, counteracting an ERG/PRC2 interaction that represses targets bound without ERK. In contrast, AKT activation alters the ERG cistrome, allowing ERG to promote luminal cell fates.

Importantly, our findings indicate that while ERG redistribution by AKT and ERG-mediated transcriptional repression by PRC2 are necessary for luminal fates, these processes are dispensable for ERG-mediated tumor formation. Therefore, our findings suggest that efforts to inhibit ERG function in prostate cancer should focus on the role of ERG in transcriptional activation, which is required for tumor-promoting activities both in the presence and absence of AKT activation.

While it is well established that PTEN deletion or PI3K activating mutations can co-operate with ERG in tumorigenesis, the role of the Ras/ERK pathway in ERG mediated tumorigenesis is less well understood. Our previous studies indicate that, due to a relatively high-affinity interaction between ERG and ERK, the Ras/ERK pathway does not need to be pathologically activated for ERG to be phosphorylated in prostate cells [32]. This is consistent with a general lack of RAS-activating mutations in prostate cancer [41]. Our data indicate that ERK is preferentially recruited to a subset of ERG binding sites, and this recruitment is what allows ERG to activate a subset of target genes. This mechanism is similar to previous findings that differential ERK recruitment can specify the repression/activation function of the ETS factor ELK1 [35]. Sites of ERG/ERK co-binding were enriched for composite ETS/AP1 sites. AP-1 factors are also known to be phosphorylated and regulated by ERK2 [42] and in interaction between ERK and AP-1 could bias ERK binding to chromatin in co-operation with ERG. ETS/AP1 composite sites are enriched in the regulatory regions of genes involved in cell migration and EMT [36], and thus ERG/ERK co-binding to these sites could be responsible for these phenotypes when ERG is expressed in RWPE1 cells.

Our findings indicate that activation of AKT allows expansion of the ERG cistrome. PTEN deletion and activation of AKT has previously been demonstrated to play a role in regulating ERG transcriptional activity through eviction of FOXO1 from the nucleus, a repressor of ERG transcription [18]. It is possible that FOXO1 blocks ERG from FOX/ETS sites and this could explain the increased binding of ERG to FOX/ETS sites in cells with constitutively active AKT. Composite FOX/ETS sites have previously been demonstrated to be critical regulators of endothelial-specific gene expression [43], which is a key normal function of ERG. Active AKT signaling also promoted novel ERG binding to GGAA repeats. GGAA repeats are critical regulatory elements in Ewing's Sarcoma, a disease driven by a translocation of the *EWSR1* gene to the DNA binding domain of ERG or FLI1, ERG's closest homolog in the ETS family [44]. Our data suggests that AKT activation could promote regulation of GGAA microsatellites in prostate cancer and could potentially aid in promoting tumorigenesis in the prostate. Although active AKT signaling promoted binding to these sites, the mechanism by which AKT promotes ERG's ability to bind these de novo sites remains unclear and warrants further investigation.

ERG can promote epithelial to mesenchymal transition [14,16,17] and conversely it can promote prostate epithelial differentiation through the basal to luminal transition [8,9,10]. The basal to luminal transition is thought to be a critical step in prostate cancer as the majority of prostate adenocarcinomas are characterized by a loss of basal cells and expansion of luminal cells [45]. The RWPE1 cell line is derived from basal epithelial prostate cells and expresses basal cell markers cytokeratins 8 and 18 [40]. Our data indicates that PI3K/AKT signaling alone in RWPE1 cells promotes further epithelial differentiation and opposes EMT, while ERG expression alone has an opposite role, promoting EMT. Strikingly, the combination of active AKT and ERG expression together in these cells cooperated to promote luminal epithelial differentiation. However, ERG S96E robustly promoted tumor formation without luminal epithelial differentiation, suggesting that ERG's ability to promote luminal fates is not the key to ERG's oncogenic function.

These findings have important implications for the roles of transcriptional activation and repression by ERG. We utilized mutations in the S96 phosphorylation site of ERG that can separate transcriptional activation and repression functions. Our data both here and in our previous study [21] indicate that ERG S96A biases ERG function towards repression, while ERG S96E can only activate. Neither ERG S96E, nor ERG S96A could promote luminal differentiation in the presence of AKT, indicating that both activating and repressing ERG functions are required for this process. This is consistent with the recent finding that direct repression of TP63 is required for ERG to promote luminal fates [8]. However, ERG S96E could promote tumor formation in the absence of activated AKT, indicating that ERG repressive function is somehow inhibiting its ability to promote tumor formation in the absence of a second oncogenic hit. This finding, coupled with the result that ERG S96A cannot promote tumor formation even in the presence of activated AKT, indicates that the best therapeutic strategy for targeting ERG function in prostate cancer is to inhibit its transcriptional activation function.

## Materials and methods

### Ethics statement

All animal protocols described in this study were approved by the Institutional Animal Care and Use Committee (IACUC) at the Indiana University School of Medicine under protocol 11324.

### Cell culture, retroviral expression, migration, and clonogenic growth assays

RWPE1 (ATCC) were cultured in K-SFM with .05 mg/ml BPE, 5 ng/ml EGF and Penicillin and Streptomycin. VCaP cells were cultured in DMEM with 5% FBS and Penicillin and Streptomycin. VCaP cells used for ChIP were treated with either DMSO or 1uM LY294002 on day 1, day 3, and day 6, followed by harvesting on day 7. *ERG* and phospho-mutants were expressed in pQCXIH as previously described [21]. Myr-Akt was obtained from Addgene 15294 and pBABE-puro was used as vector control. HEK-293T cells were co-transfected with retroviral expression plasmids, PN8E gagpol delta5, pN8E VSV-G plasmids to create retroviruses. Retroviruses were added to RWPE1 cells with 10mg/ml polybrene for 4 h followed by addition of growth media. After 24 hours cells were maintained under hygromycin (250 ug/ml) and puromycin (20 ug/ml). Migration assays were performed as previously described [15]. Briefly, 5 x $10^4$ cells were plated in 8 micron Boyden chamber (BD Biosciences) in K-SFM without supplements and allowed to migrate for 72 hours to normal growth media before staining and counting. Clonogenic growth assays were performed by plating 1,000 cells per well in a six-well plate and allowed to grow for 10 days before staining and colony quantification using GeneTools Syngene software.

### Mouse xenograft experiments

Experiments were performed as previously described [19]. In short, 2 x $10^6$ RWPE1 cells were combined with equal volume Matrigel and .5 x $10^6$ patient derived cancer-associated fibroblasts or DLP derived stromal cells from INK4A null mice and injected into the flank of nude mice. Weekly tumor measurements were taken over the course of 8 weeks using calipers and were calculated using the formula: volume = length x width x height x 0.5236.

### Immunoblot and ChIP

Whole-cell extracts of equivalent cell number were run on 12.5% SDS-PAGE, transferred to nitrocellulose membrane and blocked with 5% milk in TBST. Proteins were detected with

anti-Flag M2 (Sigma Life Science), Tubulin (Sigma T9026), Phospo-AKT S473 (CST D9E), ERG (CM421 Biocare) Histone 3 (CST D1H2), FOXA1 (Abcam ab23738), AR, TP63(Invitrogen 10H7L17) (Abcam ab108341), and ERG P-S215 (custom antibody). ChIP of indicated proteins was performed as previously described [15] with the exception of the use of anti-mouse or anti-rabbit Dynabeads (ThermoFischer) using antibodies ERK2 (D-2 SCBT), EZH2 (D2C9 CST), and H3K27me3 (CST C36B11), ERG (CM421 Biocare).

## ChIP-Seq and analysis

ChIP-Seq library preparation was performed as previously described [36]. Briefly, ChIP samples were pooled from three independent ChIP experiments and were sheared to ~150 bp fragments using a Diagenode BioRuptor. Fragments were end repaired and TruSeq (Illumina) adapters were ligated followed by PCR amplification, size selection from a 2% agarose gel, and bead cleanup using Ampure XP beads. Libraries were multiplexed and sequenced with a NextSeq75 – High Output on an Illumina NextSeq500. All samples were sequenced single end. Reads were aligned with Bowtie2 (hg19), Black List Regions were removed using bedtools, and duplicates were removed using Picard-2.14.0. Peaks were called with MACS2 using p-value .001. Metagene plots and heatmaps were generated with NGSPlot, Motifs were discovered using RSAT (http://rsat.sb-roscoff.fr/), USeq FindNeighboringGene tool was used to find nearest genes, and Metascape (Metascape.org) identified Gene Ontologies. All ChIP-seq data is available from the gene expression omnibus (https://www.ncbi.nlm.nih.gov/geo/) under accession number GSE164859.

## ATAC-sequencing

ATAC-Sequencing libraries were conducted using ActiveMotif ATAC-Sequencing kit. Briefly, RWPE1 cells were counted and 50,000 cells were spun down, washed with PBS, and lysed with ActiveMotif ATAC lysis buffer followed by tagmentation for 30 minutes. Tagmented DNA was purified, PCR amplified for 10 rounds, and bead cleaned using SPRI beads. Sequenced DNA was aligned using Bowtie2 (hg19), duplicates were removed using Picard-2.14.0, and peaks were called with MACS2 using the broad peaks setting. Venn Diagram was generated using Galaxy Cistrome (cistrome.org), gene tracks were viewed using IGV2.6.2, and heatmaps were generated using NGSPlot.

## Immunoprecipitation, cell fractionation, and mass spectrometry

RWPE1 and VCaP were lysed with NP-40 lysis buffer, sonicated using a probe sonicator, and spun down. ERG antibody was incubated with supernatant overnight followed by 2 hr addition of anti-mouse Dynabeads. Beads were washed 4 times with lysis buffer, 4X lamelli buffer was added, and samples were run on 12.5% SDS-PAGE gels followed by size selection into 1% acetic acid. Mass Spectrometry was conducted and data was analyzed using Scaffold 4. Nuclear/cytoplasmic fractionation was conducted by lysing cells in Cell Lysis buffer(50mM Hepes, 1mM EDTA, .5mM EGTA, 140 mM NaCl, 10% Glycerol, .5% NP40, protease inhibitors), nuclei were pelleted, washed 1X with Wash Buffer (10mM Tris, 1mM EDTA, .5mM EGTA, 200mM NaCl, protease inhibitor) and then lysed with NP40 lysis buffer(150mM NaCl, 1% NP-40, 50mM Tris). Nuclear and cytoplasmic fractionations were boiled in 4X lamelli buffer and were then subjected to western blotting.

## RNA extraction, RNA-Seq, and analysis

RNA was extracted using Qiagen RNA extraction kits, polyA selected using Dynabeads Oligo (dT) (Invitrogen 61002), reverse transcribed and libraries were prepared as previously

described (Kedage et al. 2017). Reads were aligned using TopHat2 and differential gene expression was determined using Cuffdiff. GSEA was conducted using pre-ranked lists generated by multiplying log2(FC)*(1/P-value). Heatmaps were generated used Morpheus (https://software.broadinstitute.org/morpheus/). Volcano plots were generated using Galaxy VolcanoPlot tool (https://usegalaxy.org/). Differential Gene Expression matrix was made with cummeRbund. All RNA-seq data is available from the gene expression omnibus (https://www.ncbi.nlm.nih.gov/geo/) under accession number GSE164859.

## Immunohistochemistry

Heat-fixed paraffin-embedded sections were processed for immunofluorescence analysis as previously described [46,47]. Slides were incubated overnight at 4˚C with one of the following antibody combinations diluted in blocking buffer (10% normal donkey serum, 1% bovine serum albumin in PBS-T): mouse anti-pan-cytokeratin at 1:150 (Cell Signaling 4545) and rabbit anti-vimentin at 1:100 (Cell Signaling 5741); rabbit anti-Keratin 5 at 1:1000 (CAT#: PRB-160P, Covance), or mouse anti-cytokeratin 8 (CAT#: NB120-9287, Novus). After sections were washed with PBS-T, Invitrogen Alexa Fluor 488 or 594 anti-mouse or anti-rabbit IgG secondary antibodies at 1:150 were applied for 1 hour at room temperature in the dark. After serial washes with PBS-T, sections were incubated with 20 µg/mL Hoechst 33342 in PBS for 10 min to visualize nuclei. Coverslips were mounted using PermaFluor Mountant media (CAT#: TA-030-FM, Thermo). Slides were viewed using a Leica 6000 epifluorescent microscope and digital images were captured using Leica LAS software.

## Supporting information

**S1 Fig. ERG S96E increases RWPE1 clonogenic growth and promotes tumor formation in a mouse xenograft model.** (A) Representative images of migration assays quantified in Fig 1B. (B) Representative images of clonogenic growth quantified in Fig 1C. (C) Weekly xenograft tumor volume measurements of RWPE1 and DLP derived INK4a null cells injected into the flanks of nude mice.
(TIF)

**S2 Fig. Ras/ERK signaling occurs at ETS/AP1 genomic locations to regulate cell migration.** (A) Mass Spectrum identifying phosphorylation of ERG S96 in VCaP and peptide coverage of ERG protein with PTMS highlighted in green. (B) Mass Spectrum identifying phosphorylation of ERG S96 in RWPE1 cells and peptide coverage of ERG protein. (C) Motif analysis of ERK binding sites in RWPE1 and RWPE1-ERG cells. (D) ERK ChIP-Seq coverage centered on regions bound by ERK in both datasets, and not bound by ERG.
(TIF)

**S3 Fig. Ras/ERK signaling opposes PRC2 repression at ERG binding sites.** (A) Heatmap of genes nearest ERG bound regions in RWPE1 cells comparing RNA-Seq Log2(FC) of gene expression between RWPE1, RWPE1-ERG, and RWPE1-ERG S96E. (B) Gene Ontology analysis of nearest genes to ERG bound regions in RWPE1 cells that are significantly activated in RWPE1 ERG S96E compared to RWPE1 ERG.
(TIF)

**S4 Fig. Active PI3K/AKT signaling alters the ERG cistrome and transcriptome.** (A) Kmeans clustered heatmap of ERG ChIP-Seq data from indicated lines centered on a combined peak list merged from all four datasets. (B) ChIP-Seq called peaks of ERG in VCaP cells treated with 1uM LY294002 or vehicle for one week and (C) gene ontologies of closest ERG bound gene; ERG co-bound (purple), vehile treated unique (red), and ERG LY294002 treated

unique (blue). (D) Gene tracks of ERG ChIP-Seq in VCaP cells treated with vehile (red) or LY294002 (blue).
(TIF)

**S5 Fig. Active PI3K/AKT signaling alters the ERG transcriptome.** Heatmap Comparison of RNA-Seq data depicting significantly differentially regulated genes between the RWPE1 cell lines indicated.
(TIF)

## Acknowledgments

We would like to thank the Indiana University Center for Genomics and Bioinformatics for DNA sequencing and the Indiana University School of Medicine Proteomics Facility for Mass Spectrometry.

## Author Contributions

**Conceptualization:** Brady G. Strittmatter, Peter C. Hollenhorst.

**Formal analysis:** Brady G. Strittmatter, Peter C. Hollenhorst.

**Funding acquisition:** Peter C. Hollenhorst.

**Investigation:** Brady G. Strittmatter, Travis J. Jerde.

**Methodology:** Brady G. Strittmatter, Travis J. Jerde, Peter C. Hollenhorst.

**Supervision:** Peter C. Hollenhorst.

**Writing – original draft:** Brady G. Strittmatter, Peter C. Hollenhorst.

**Writing – review & editing:** Brady G. Strittmatter, Travis J. Jerde, Peter C. Hollenhorst.

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
