## [Decision Letter · Decision Letter 0]

22 Mar 2021

Dear Dr Hollenhorst,

Thank you very much for submitting your Research Article entitled 'Ras/ERK and PI3K/AKT signaling differentially regulate oncogenic ERG mediated transcription in prostate cells' to PLOS Genetics.

The manuscript was fully evaluated at the editorial level and by independent peer reviewers. The reviewers appreciated the attention to an important problem, but raised some substantial concerns about the current manuscript. Based on the reviews, we will not be able to accept this version of the manuscript, but we would be willing to review a much-revised version. We cannot, of course, promise publication at that time.

If you decide to revise the manuscript for further consideration at PLOS Genetics, please aim to resubmit within the next 60 days, unless it will take extra time to address the concerns of the reviewers, in which case we would appreciate an expected resubmission date by email to plosgenetics@plos.org.

[LINK]

We are sorry that we cannot be more positive about your manuscript at this stage. Please do not hesitate to contact us if you have any concerns or questions.

Yours sincerely,

Charles L. Sawyers

Guest Editor

PLOS Genetics

David Kwiatkowski

Section Editor: Cancer Genetics

PLOS Genetics

Your manuscript has been reviewed by three expert referees, all of whom find the work of interest, particularly the effects of AKT on the ERG cistrome and induction of luminal gene expression (such as AR and FOXA1). However, several concerns are raised that must be addressed in a revision, specifically:

1) extension of the findings to another model besides RWPE cells, such as VCAP

2) mechanistic insight into how AKT mediates its effects on ERG and induction of luminal gene expression

3) more evidence of luminal lineage specification

In addition, there are significant concerns with the quality of the ERG ChIP data identified by reviewer 2 that must be addressed.

Reviewer's Responses to Questions

**Comments to the Authors:**

Reviewer #1: In the present study the authors aimed to assess the impact of ERK and AKT pathways on ERG function in chromatin binding, gene repression/transactivation and tumorigenesis. They found that ERK co-localized with ERG in the nucleus and in a subset of ERG target genes and ERK phosphorylation of ERG at S96 results in loss of ERG association with EZH2 and H3K27me3 repression histone mark. They further showed that S96E is important for ERG mediated tumorigenesis, but this effect is unlikely caused by changes in ERG cistrome. In contrast, AKT expression largely induces ERG cistrome reprogramming and most strikingly, co-expression of AKT with ERG converts AR-null RWPE1 cells into AR-positive luminal cells, although the authors point out that the underlying mechanism is not entirely clear.

Overall, it is a very exciting study and data are quite convincing. The data are well interpreted and the conclusion is supported by the data presented. Only a few minor issues need to be addressed.

The data shown in Figure 3A is very intriguing. It is important to determine whether S96E phosphomimetic mutant induced decrease in EZH2 binding and H3K27me3 levels at ERG target promoters and enhancers is statistically significant.

It would be more informative if top 5 or 10 genes activated by S96E are marked in Figure 3C.

In Line145 in the phase of “S103 in Uniprot isoform 1”, it should be S103 in isoform 2. Since ERG isoform 2 is the one most frequently detected in TMPRSS2-ERG fusions in patients and the numbering of phosphorylation site is likely based on isoform 1, it is important for authors to indicate in the upfront of the Results section that ERG isoform 1 was used in the current study and briefly mention the differences between ERG isoforms, especially with isoform 2.

Figure 2B in Line 298 should be Figure 4B.

In line 333 and 335, it seems the description of the results does not match to the figures cited.

The paper reported by Blee et al. (Clin Cancer Res 2018) should be cited in the context of discussion of the role of ERG in promoting luminal epithelial phenotype, especially in cells with PTEN loss.

Reviewer #2: Prostate cancer is characterized by ERG overexpression through translocation in ~ half of all cases. There is co-occurrence with PTEN loss and several model systems have recapitulated cooperativity between PTEN loss and ERG overexpression in tumorigenesis. This manuscript addresses the role of PI3K/AKT as well as Ras/ERK in ERG function.

Overall, the manuscript tells two separate stories. The first is the role of ERG phosphorylation on S96 in tumorigenesis and the second is the role of AKT activation on ERG activation and binding. Much of the data in the first story has been previously published in Kedage et. al. JBC 2017, including that ERG S94E by itself stimulates migration, gene activation, and decreased PRC2 suppression at sites. The second story is more novel and potentially impactful. It is quite interesting that adding of both ERG and myrAKT in RWPE cells causes expression of FOXA1 and AR. However, the data presented is not well flushed out and lacks quality controls.

Major Comments:

1. I am quite worried about the quality of ChIP-seq that has led to the conclusions in the manuscript. ERK ChIP peaks in Fig 1C and 1E are very broad, though that may be it’s true biology. More concerning are ERG peaks in Fig. 4A, which should be sharp but seem very broad. The authors did not make the raw files for reviewers to examine. However, this reviewed did go into the Cistrome.org public repository where ERG ChIP-seq in RWPE cells were deposited from this lab’s prior work as well as from the lab (Kedage et. al., Cell Rep 2016) of David Rickman (Rickman et. al. PNAS 2012). It is worrisome that the deposited samples from Kedage et. al did not meet quality control of cistrome.org for # peaks, peaks at known open regions, etc. For authors, I’ve attached a PDF file, 1st page of the DUSP6 locus with known ERG peaks in all lineages. Top two tracks are VCAP and Rickman RWPE and below are tracks from Kedage et. al. The next 3 pages are QC metrics from cistrome.org of Rickman RWPE and Kedage RWPE. For resubmission, authors need to submit all the QC metrics of their ChIP-seq as well as bed files for where the peaks are and bigwig files.

2. Essentially all the work has been done in RWPE cells, and its relevance is questionable. It is well known that VCAP ERG cistrome and transcription is very distinct from RWPE. The findings should be recapitulated in VCAP cells.

3. PTEN loss followed by PIK3CA and PIK3R1 mutations are clinically relevant alterations of the PI3K pathway. Can authors recapitulate at least the phonotype using these alterations?

4. The increase in ERG binding sites in RWPE expressing mAKT is striking and the most novel and impactful part of the paper. This finding is in contrast to prior mouse models (Chen et. al, Nat Gen). There is no mechanistic insight. Are the new sites a reflection of opening of chromatin and change in the open chromatin landscape by mAKT or increased ERG binding to established enhancer/promoters? In RWPE with mAKT and also in VCAP’s, does ERG cistrome shrink upon inhibition of AKT?

5. Specifically, can ERG phosphorylation be inhibited by novel ERK kinase inhibitors and does that change ERG mediated cistrome/transcriptome

6. In fig 1D-E, In vivo tumor formation was done with RWPE cells and immortalized MEFs or CAFs. Authors need to show histology to confirm that tumors are epithelial and not sarcomas from the fibroblasts.

7. In Fig 2E, authors need to show ERK phosphorylation is not changed by ERG expression (ERG can induce feedback genes that inhibit ERG phosphorylation) and that ERK ChIP-seq signal at non-ERG sites is unchanged (i.e., change is specific to ERG sites).

8. Fig 5A is hard to interpret for overall gene expression.. A unsupervised clustering and correlation clustering should also be shown.

9. (Jerde et al. 2009) not in the reference list

Reviewer #3: The review is uploaded as an attachment

**Have all data underlying the figures and results presented in the manuscript been provided?**

Reviewer #1: Yes

Reviewer #2: **No: **Data needs to be made public and analyzed data needs to be made assessible to reviewers.

Reviewer #3: Yes

PLOS authors have the option to publish the peer review history of their article (what does this mean?). If published, this will include your full peer review and any attached files.

Reviewer #1: **Yes: **Haojie Huang

Reviewer #2: No

Reviewer #3: No

---

## [Decision Letter · Decision Letter 1]

19 Jun 2021

Dear Dr Hollenhorst,

Thank you very much for submitting your Research Article entitled 'Ras/ERK and PI3K/AKT signaling differentially regulate oncogenic ERG mediated transcription in prostate cells' to PLOS Genetics.

The manuscript was fully evaluated at the editorial level and by independent peer reviewers. The reviewers appreciated the attention to an important topic but identified some concerns that we ask you address in a revised manuscript

We therefore ask you to modify the manuscript according to the review recommendations. Your revisions should address the specific points made by each reviewer.

[LINK]

Yours sincerely,

Charles L. Sawyers

Guest Editor

PLOS Genetics

David Kwiatkowski

Section Editor: Cancer Genetics

PLOS Genetics

The manuscript is much improved but we are still concerned about the quality of the ERG ChIP and whether the conclusion about ERG peaks in mAKT cells could be due to differences compared to your earlier 2017 paper. We ask that you conduct the analysis described by reviewer 2 to confirm the conclusion from analysis of mAKT cells.

Reviewer's Responses to Questions

**Comments to the Authors:**

Reviewer #2: This revised manuscript is significantly improved and addresses most of my concerns. My only remaining concern rests on the ERG ChIP-seq (Fig 4 and Supplementary Fig 4). This is the basis of expanded ERG cistrome when AKT/PI3K is activated. The use old previously obtained ChIP-seq data from prior publication (ERG and ERG S96E) as a comparator to new ChIP-seq data (ERG + mAKT) done at different time with different cells is not ideal. Ideally, current best practice would call for duplicate ChIP-seq done in parallel. However, another type of analysis that would be satisfactory would be to

1. Use 4 datasets of RWPE-ERG (Kedage 2017), RWPE-ERGS96E (Kedage 2017), RWPE-ERG from Rickman PNAS, and RWPE-ERG-mAKT (this paper).

2. Call peaks. Merge peaks and show merge statistics

3. show peaks of all 4 datasets of teh merged bed file, cluster by k-means to highlight specific peaks in each dataset.

**Have all data underlying the figures and results presented in the manuscript been provided?**

Reviewer #2: Yes

PLOS authors have the option to publish the peer review history of their article (what does this mean?). If published, this will include your full peer review and any attached files.

Reviewer #2: No

---

## [Editor Report · Decision Letter 2]

10 Jul 2021

Dear Dr Hollenhorst,

Thank you for addressing the remaining concerns about the ERG ChIP data with the new analysis in Supplemental Figure 4A.  We are pleased to inform you that your manuscript entitled "Ras/ERK and PI3K/AKT signaling differentially regulate oncogenic ERG mediated transcription in prostate cells" has been editorially accepted for publication in PLOS Genetics. Congratulations!

Yours sincerely,

Charles L. Sawyers

Guest Editor

PLOS Genetics

David Kwiatkowski

Section Editor: Cancer Genetics

PLOS Genetics

Comments from the reviewers (if applicable):

All remaining concerns have been addressed.

**Data Deposition**

http://datadryad.org/submit?journalID=pgenetics&manu=PGENETICS-D-21-00143R2

**Press Queries**

---

## [Editor Report · Acceptance letter]

22 Jul 2021

PGENETICS-D-21-00143R2 

Ras/ERK and PI3K/AKT signaling differentially regulate oncogenic ERG mediated transcription in prostate cells 

Dear Dr Hollenhorst, 

We are pleased to inform you that your manuscript entitled "Ras/ERK and PI3K/AKT signaling differentially regulate oncogenic ERG mediated transcription in prostate cells" has been formally accepted for publication in PLOS Genetics! Your manuscript is now with our production department and you will be notified of the publication date in due course.

With kind regards,

Katalin Szabo

PLOS Genetics

On behalf of:
